# Towards Biologically Plausible Learning by Stacking Circular Autoencoders

Anonymous Full Paper
Submission 14

## Abstract

Training deep neural networks in biological systems is faced with major challenges such as scarce labeled data and obstacles for propagating error signals in the absence of symmetric connections. We introduce Tourbillon, a new architecture that uses circular autoencoders trained with various recirculation algorithms in a self-supervised mode, with an optional top layer for classification or regression. Tourbillon is designed to address biological learning constraints rather than enhance existing engineering applications. Preliminary experiments on small benchmark datasets (MNIST, Fashion MNIST, CIFAR-10) show that Tourbillon performs comparably to models trained with backpropagation and may outperform other biologically plausible approaches. The code and models are available at https://anonymous.4open.science/r/Circular-Learning-4E1F.

## 1 Introduction

Decades of machine learning have taught us that gradient descent is the sole effective optimization method in high-dimensional spaces. Other strategies, like random search, are bound to fail. Backpropagation, the algorithm behind gradient computation in artificial neural networks, has been incredibly successful. It powers advancements in Artificial Intelligence, from protein folding (e.g., AlphaFold [1]) to natural language understanding and generation (e.g., GPT-4 [2, 3]). Backpropagation efficiently computes the gradient in a network with $W$ weights using $O(W)$ operations. Considering that at least $O(W)$ operations are necessary to adjust $W$ synapses, backpropagation demonstrates optimal efficiency. Consequently, if learning is viewed as an optimization problem in a high-dimensional space of synaptic weights, this suggests that the brain likely employs learning algorithms based on gradient computation, either exact or approximate. Yet there are several well-known reasons in the literature why backpropagation is implausible in biological systems [4, 5, 6, 7, 8, 9]. Thus, in short, we hypothesize that biological systems must strive to approximate gradient descent methods without being able to compute exact gradients by backpropagation. Here, we set out to propose a plausible strategy for achieving this goal.

Let us first briefly enumerate some of the major reasons why backpropagation is not plausible in biological neural networks: **1) Symmetry of Connections** (weight transport): Backpropagation requires precisely symmetric connections between the forward and backward passes. This constraint cannot be satisfied in a biological neural system and might be hard to realize in some physical neural systems. **2) Forward Nonlinearities** (F prime): Backpropagation relies on the exact memory of forward pass nonlinearities (e.g., activation functions) to compute weight updates. This is not supported in biological or physical neural systems. **3) Locality**: In a biological neural system, the learning rule for adjusting synaptic weights must be local, i.e. it must rely solely on variables available locally, both in space (spatial locality) and time (temporal locality), at each synapse. **4) Clocked Computation**: In backpropagation, the forward and backward passes are manually clocked to compute activations and update weights. In contrast, in a biological system, neurons communicate stochastically, lacking the precise clocking mechanism observed in backpropagation. **5) Labeling**: Backpropagation relies on large labeled datasets, unlike biological systems, which lack access to such data. **6) Spike**: Biological neurons use noisy spikes for communication, while artificial neurons typically use deterministic analog values. **7) Distances**: Backpropagation necessitates propagating signals over considerable neural distances in deep models, which can result in signal dilution and lead to distorted or unstable gradients. **8) Developmental Modularity**: Backpropagation in general, requires having a complete architecture in place before training can begin, which may not be realistic for biological systems undergoing development and other changes.

Several solutions have been suggested to try to address these problems, in isolation or small combinations, but no approach addresses all of them at once. Here we propose a neural architecture called Tourbillon and its training algorithms to address all the implausibility discussed above by combining different ideas, including stacked autoencoders, recirculation, and asynchronous training. We emphasize that the primary goal here is to address the obstacles listed above for biological (or neuromorphic) neural systems and not to derive a new architecture or algorithm that is practically useful for digital applications of deep learning.

## 2 Biological Plausibility

Several approaches have been proposed to address the biological implausibilities enumerated above. The most notable ones include: Feedback Alignment (FA), Difference Target Propagation (DTP), Stacked Autoencoders, and the Forward-Forward (FF) algorithm. However, each of these methods addresses only a limited subset of the biological implausibilities (Section A.1 and Table 1). Self-supervised learning, in particular stacked autoencoders, provides one way of addressing the data labeling issue. However, standard autoencoders suffer from several other issues which we now address.

**Circular Autoencoders.** In a standard feed-forward autoencoder (AE), the data itself provides the targets (self-supervised learning). The data and hence the targets are available in the input layer. However, they are not available in the output layer, in the sense that they are not physically local (spatial-locality) to the output layer. This problem is addressed in circular autoencoders (CAE) [8] where the output layer is physically equal (or physically adjacent) to the input layer (Figure 1). With the circular layout, targets and errors can be computed at the level of the input/output layer.

**Recirculation Algorithms.** Standard backpropagation, or even FA, of these targets, would require a channel (wires) running backward from the output layer to the hidden layer. However, because of the circular layout, it is possible to use the forward connections to propagate target and error information during learning. This is the fundamental idea behind recirculation, a family of algorithms for training CAEs that do not require backward connections [10, 11, 8].

Consider a CAE with layers numbered from 0 to $L$, where 0 corresponds to the input layer. We use the index $t$ to denote different cyclic passes through the autoencoder, with the first pass indexed by $t = 0$. After the first pass, one can *locally* compute the error $T - H_L^0$, where $T$ is the target located at the input layer. This error could be used to train the top layer of the CAE by gradient descent, and then train the other layers by using a form of random backpropagation where the error signal is obtained by propagating the error $T - H_L^0$ using the forward weights of the CAE. This however requires propagating two different kinds of signals, activities, and errors, through the CAE. Thus rather than recirculating the error, a more uniform approach can be obtained by recirculating activities. If $H_i^t$ denotes the activation of layer $i$ during the forward pass indexed by $t$, the main idea behind the recirculation family of algorithms is to use $H_i^t$ as the target for the output $H_i^{t'}$ taken at a later time $t'$ to produce the post-synaptic term for the weight update. The intuition is that the data may become increasingly corrupted as it is being recycled, thus earlier pass serve as targets for later passes. Different variations can be obtained, by varying, for instance, the post- and pre-synaptic terms. In addition to the original learning rule of recirculation [8] Equation 1, we propose two variations of the learning rule, all shown in Equation 9 (b) and (c) in the Appendix.

$$\Delta W_i = \eta (H_i^t - H_i^{t'})^{post} (H_{i-1}^t)^{pre} \qquad (1)$$

These rules follow a Hebbian-product form, resembling backpropagation but with a postsynaptic recirculation error, denoted as $[H_i^0 - H_i^1]^{post}$. This error term is both spatially and temporally local, assuming that consecutive passes through the circular autoencoder fall within the proper time window. In the input layer, the vector $H_0^0$ represents the input data, including the targets for an autoencoder. The presynaptic term can be computed at different times ($t$ or $t'$) or even as the difference between the activities at two different times. Using the first form or presynaptic terms (Equation 1), the recirculation learning equation for the top layer of weights is identical to backpropagation. Although in this work we are not using spiking neurons, all learning rules described in Equation 9 are closely related to the concept of spike time-dependent synaptic plasticity (STDP) [12]. STDP Hebbian or anti-Hebbian learning rules have been proposed using the temporal derivative of the activity of the post-synaptic neuron [13] to encode error derivatives.

## 3 Tourbillon: A CAE Stack

We propose the Tourbillon architecture as a stack of circular autoencoders, capped by a classification or regression layer connecting the hidden representation of the top circular autoencoder and the output layer. Each circular autoencoder has an encoder and decoder components. The hidden layer that is shared by the encoding and decoding components is called the hinge layer. In the stack, the hinge layer of the $i$th circular autoencoder becomes the input layer of the $i + 1$th circular autoencoder (Figure 2). The Tourbillon architecture addresses the issues of target labels and spatial locality. With the recirculation algorithms, it also addresses the issues of weight transport, forward non-linearities, temporal locality, and distances. Using a novel training algorithm, we set out to address issues of clocking and modularity.

**Sequential Training.** In sequential training, the CAEs are trained separately. Training of the $i$-th CAE in the stack must be completed before training of the $i + 1$-th CAE can begin. The input data to the $i + 1$-th CAE is provided by the hidden representations produced by the hinge layer of the $i$-th CAE. Finally, we can stack $N$ trained CAE, each is trained to further compress the hinge layer of its

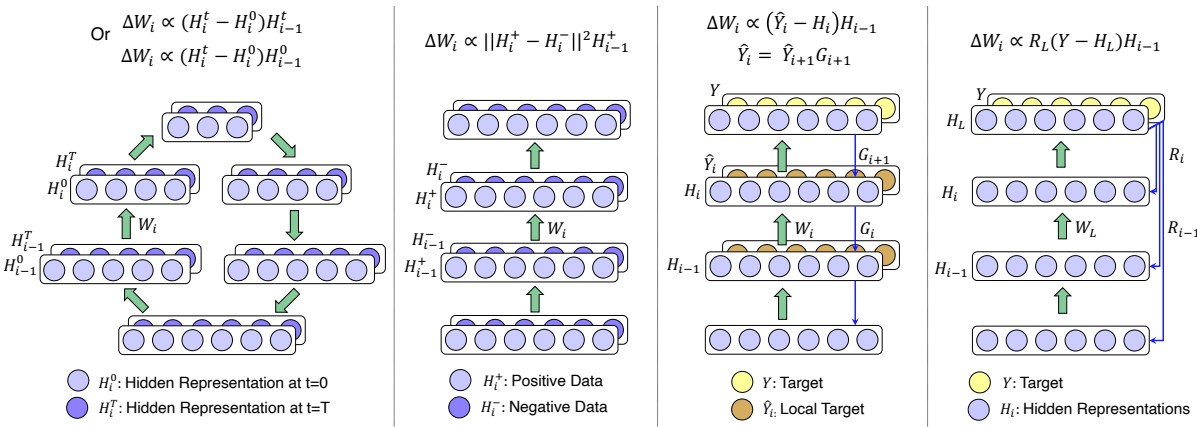

**Figure 1.** From left to right: Recirculation, forward forward, difference target propagation, (direct) feedback alignment. The learning rule for each model is written at the top of the architecture schematics.

**Table 1.** A comparison of physical plausibility between different neural architectures from a biological standpoint. ✘, ⊗, and ✔ correspond to no plausibility, partial plausibility, and full plausibility, respectively.

|  | W Transport | F prime | Locality | Clocked | Labeling | Spike | Distance | Modular |
|---|---|---|---|---|---|---|---|---|
| Backpropagation | ✘ | ✘ | ✘ | ✘ | ✘ | ✘ | ✘ | ✘ |
| Feedback Alignment (FA) | ✔ | ✔ | ✘ | ✘ | ✘ | ✘ | ✘ | ✘ |
| Direct Feedback Alignment (DFA) | ✔ | ✔ | ⊗ | ✘ | ✘ | ✘ | ✔ | ✘ |
| Difference Target Propagation (DTP) | ⊗ | ✘ | ✘ | ✘ | ⊗ | ✘ | ✔ | ✘ |
| Stacked Autoencoders | ✔ | ✔ | ⊗ | ✘ | ✔ | ✘ | ✔ | ⊗ |
| Forward Forward (FF) | ✔ | ✔ | ✔ | ✘ | ⊗ | ✘ | ✔ | ✘ |
| **Tourbillon** | ✔ | ✔ | ✔ | ✔ | ⊗ | ✘ | ✔ | ✔ |

predecessor CAE. These trained CAEs as the building blocks of tourbillon must be trained from the bottom to the top. So each autoencoder would be able to generate the training data for its successor building blocks. This is explained in Equation 3 where $E$ and $D$ represent the encoder and decoder characterized by a deep neural network trained with recirculation. $x$ is the original training data (e.g. MNIST) and $H$ shows the hidden representation of the circular autoencoder.

$$H_1 = E_1(x_1), \hat{x}_1 = D_1(H_1), \quad (2)$$

$$H_2 = E_2(H_1), \hat{H}_1 = D_2(H_2) \quad (3)$$

**Asynchronous Training.** The sequential training algorithm is the standard method for training a stack of autoencoders by fully training the first AE, then the second one, and so on. This requires a high degree of orchestration. Moreover, sequential training does not fully support modularity, as training each CAE depends on the prior training of all preceding CAEs. To remove the need for clocked orchestration and increase biological plausibility via modularity, we introduce the asynchronous training algorithm where all the CAEs are trained simultaneously and asynchronously. The main idea behind asynchronous training is to train the weights of one CAE of the stack that is randomly chosen. In this case, each CAE can be viewed as a "spinning wheel" and these wheels can spin independently of each other. At any random time, a CAE may elect to recirculate whatever happens to be in its input layer (either random data, older data, or the data at the hinge layer activated by previous CAEs) and adapt its synapses accordingly. Further details and a precise algorithm on asynchronous training are given in the Appendix.

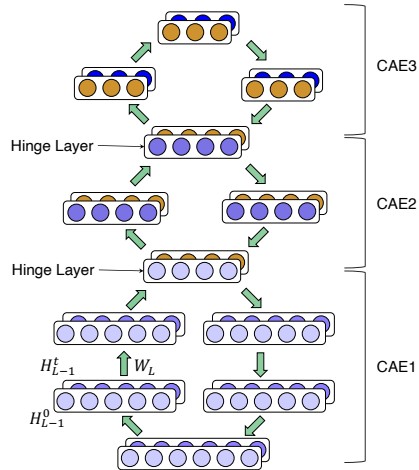

**Figure 2.** Tourbillon architecture with a stack of three circular autoencoders (CAE) trained by recirculation.

## 4 Experiments and Results

We begin by training CAEs to investigate the effects of various parameters, including the number of cycles (between $t$ and $t'$), and the CAE size (i.e.,

the number of hidden layers in the CAE except the input and output layers). We use the learning rule in Equation 1 in training CAEs, however a discussion on the effect of different learning rules (Equation 9) is presented in Section A.2.1. Then, using the best set of these parameters, we develop and train several Tourbillon architectures, both with and without a top classifier layer, using different stack depths and training algorithms. The goal of these experiments is not to outperform existing deep learning models but to show that Tourbillon architectures can learn complex tasks while satisfying the plausibility constraints. Similar to recently proposed plausible architectures [14], we use relatively small datasets and models, leaving the scaling up to future studies. Details on the hyperparameters, hardware, and CAE implementation can be found in the Appendix and the GitHub repository.

## 4.1 Training Tourbillon CAEs

We train CAEs using the learning rule in Equation 1. We optimize each architecture using a mean-squared reconstruction loss. In all experiments, we use symmetric CAEs, where the number of hidden layers in the encoder and decoder are equal. To satisfy distance plausibility, we use CAEs with a small number of hidden layers. Additionally, to maintain the temporal locality of the variables, we limit the number of cycles (difference between $t$ and $t'$) to one, two, and three.

We train CAEs with fully connected layers for the MNIST and Fashion MNIST datasets and convolutional layers for the CIFAR-10 dataset. Table 2 displays the reconstruction losses on the test datasets. Notably, using a CAE size of one and one cycle ($t = 0$ and $t' = 1$) yields the lowest testing loss. This also corresponds to the highest level of spatial and temporal locality of variables.

Using the best values for the CAE size and number of cycles, we further show the viability of training CAEs with the learning rule above. We compare the mean-squared loss of the trained CAE with the same autoencoder trained with backpropagation (BP) and feedback alignment (FA). Figure 3 shows the training and test error curves for the MNIST and Fashion MNIST datasets. Figure 5 (first row) shows the trajectories of training and test error for CAEs with convolutional layers and similar autoencoders trained with BP and FA. Additionally, randomly sampled MNIST images are reconstructed using three models shown in Figure A.3 with the average reconstruction loss. The results presented in these figures show that recirculation outperforms FA across different datasets and architectures both in training and reconstruction loss. Recirculation also shows a reconstruction loss very close to that of BP in both fully-connected and convolutional architectures (0.001% relative error). Detailed training

parameters for each CAE are given in the Appendix.

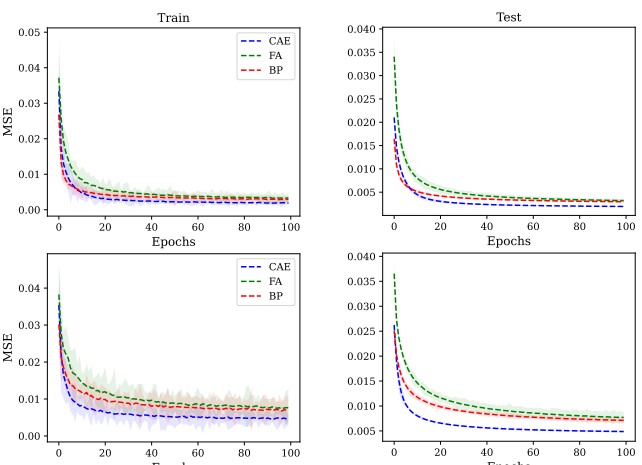

**Figure 3.** Train and test loss of three equivalent autoencoders. The first row shows the performance of the models for the MNIST dataset, and the second row shows the performance for the Fashion MNIST dataset.

## 4.2 Stacking CAEs with Various Stack Depths and Training Algorithms

We construct stacks of two, three, and four compressive CAEs and train them using sequential and asynchronous training algorithms. Table 3 demonstrates the reconstruction errors of the stacks, indicating the effect of stack depth and training algorithm applied to each CAE. Detailed parameters, such as CAE dimensions, batch size, and learning rate, are given in the Appendix.

During training, we found the asynchronous training algorithm to be sensitive to learning rate changes. Experimenting with different schedules for each CAE in the stack reveals that decreasing the learning rates from the bottom to the top layers is crucial. Despite the algorithm's inherent randomness, it achieves an identical test reconstruction error to that of Tourbillon trained sequentially. Thus, asynchronous training increases the model's biological plausibility while maintaining its performance.

After training the stacks, we add a top classifier layer for MNIST, Fashion MNIST, and CIFAR-10 datasets. We conduct classification experiments to compare the performance of different Tourbillons with neural networks of similar, but sequential architecture trained using BP, FA, and a stack of autoencoders (SAEs) with identical parameters to CAEs but trained with backpropagation.

Three different Tourbillons are considered in this experiment: (1) Tourbillon with the sequential learning algorithm (Tourbillon-seq); (2) Tourbillon with the asynchronous learning algorithm (Tourbillon-asynch); (3) and Tourbillon with the asynchronous learning algorithm that only uses 10% of the labeled

**Table 2.** The test mean-squared reconstruction loss of CAEs with different cycles and CAE sizes trained on MNIST, Fashion MNIST, and CIFAR-10. Each number is the mean of five distinct runs. The top results are in boldface.

| CAE size | MNIST | | | Fashion MNIST | | | CIFAR-10 | | |
|---|---|---|---|---|---|---|---|---|---|
| | Cycles | | | Cycles | | | Cycles | | |
| | 1 | 2 | 3 | 1 | 2 | 3 | 1 | 2 | 3 |
| 1 | **0.0090** | 0.0093 | 0.0099 | **0.0123** | 0.0124 | 0.0132 | 0.0013 | **0.0012** | 0.0014 |
| 3 | 0.0151 | 0.0154 | 0.0165 | 0.0204 | 0.0198 | 0.0204 | 0.0324 | 0.0323 | 0.0381 |

**Table 3.** The test mean-squared reconstruction loss of Tourbillon with different stack depths and training algorithms on MNIST, Fashion MNIST, and CIFAR-10. The top results are in boldface.

| Training Algorithm | MNIST | | | Fashion MNIST | | | CIFAR-10 | |
|---|---|---|---|---|---|---|---|---|
| | Stack Depth | | | Stack Depth | | | Stack Depth | |
| | 2 | 3 | 4 | 2 | 3 | 4 | 2 | 3 |
| Sequential | **0.009** | 0.026 | 0.027 | **0.013** | 0.031 | 0.032 | **0.023** | 0.046 |
| Asynchronous | **0.009** | 0.019 | 0.025 | 0.014 | 0.029 | 0.042 | 0.034 | 0.074 |

data to train the top classifier (Tourbillon-10%). All three models achieve a similar level of good classification accuracy. The Tourbillon-10% model is the most biologically plausible and performs on par with SAE which is less biologically plausible.

Although Tourbillon's performance falls short of BP in fully connected architectures (Figure 4), its viability is demonstrated by its trainability and competitive accuracy, achieving 92% test accuracy compared to 99% for BP. Additionally, Tourbillon matches the performance of SAE, which is trained with BP but is less biologically plausible. When using convolutional layers, Tourbillon performs comparably to BP and surpasses other less plausible models like FA and SAEs (Figure 5).

Additionally, we assess Tourbillon's reconstruction capability. Initially, we employ the trained stack of circular autoencoders to generate compressed representations of the input data. These compressed representations are then mapped onto a 2D space using the t-SNE method [15]. Figure 7 shows these 2D maps, highlighting Tourbillon's unsupervised ability to cluster elements in each class. Moreover, we utilize the trained stack of circular autoencoders with feed-forward fully connected layers to reconstruct the original input images by leveraging the decoder weights of the circular autoencoders. Figure 7 demonstrates that the Tourbillon models successfully capture the crucial information of the input images. During testing, the uncorrupted input is passed through the encoder channel to the top of the model and then back through the separate decoder channel to produce the reconstructed sample. In this scenario, each CAE except the top one decodes a representation in their hinge layer that comes from the CAE immediately above, and not from the CAE itself. This may explain why the reconstruction error seems to increase with the depth of the stack (Table 3). However, this does not imply that stacks with multiple CAEs are ineffective as we are interested in using the hidden representation at the top of the stack for classification or regression and not for reconstruction.

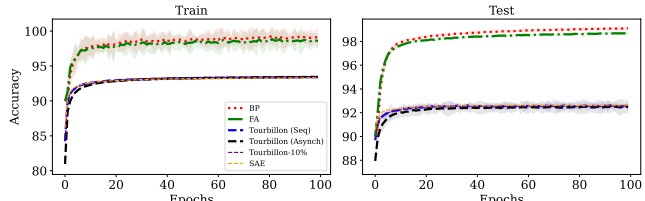

**Figure 4.** Train and test accuracy of six classifiers trained on the MNIST dataset BP, FA, Tourbillon Sequential (seq), Tourbillon Asynchronous (asynch), Tourbillon with 10% of the labeled data, and SAE. Each line corresponds to the mean of five distinct runs with the standard deviation shown as the shaded area.

## 4.3 Tourbillon Requires Less Labeled Data

A key advantage of Tourbillon is its ability to leverage unlabeled data for unsupervised training of the stack, allowing the top classifier to be trained with significantly less labeled data. To evaluate this, we use a stack of CAEs, identical to the one in the previous section, with the asynchronous training algorithm. Once the stack is trained in an unsupervised manner, we train the top classifier in a supervised manner using 10%, 25%, 50%, and 100% of the labeled data. The results for both training and testing are presented in Figure 6.

Remarkably, Tourbillon's performance remains unchanged even with only 10% of the data. This is because the stack of CAEs produces an abstract and informative representation of the input, enabling the small top-layer classifier to learn the classification task with a minimal amount of labeled data.

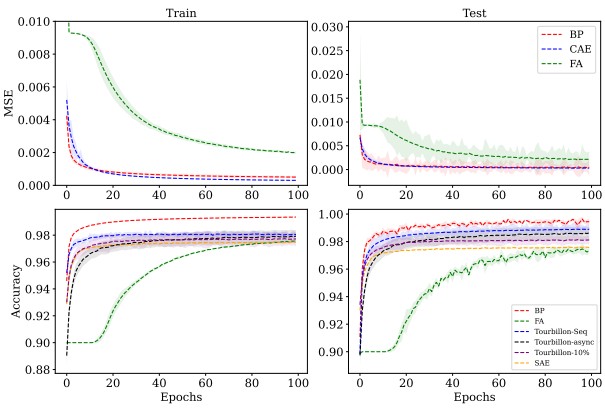

**Figure 5.** Top: The mean-squared reconstruction loss for three autoencoders trained with BP, FA, and CAE on CIFAR-10. Bottom: Train/test accuracy for six classifiers trained with BP, FA, Tourbillon-seq, Tourbillon-asynch, SAE, and Tourbillon-10% on CIFAR-10. Each trajectory is the mean of five runs, with the shaded area representing the standard deviation.

### 4.4 Conversion to Tourbillon

Finally, we introduce an algorithm designed to convert a traditional neural network with a sequential layout into its Tourbillon version.

By applying this algorithm to a layered feed-forward architecture with no skip connections, we can create a more biologically plausible counterpart that preserves the original model's functionality and core structure. The Tourbillon version has roughly twice as many parameters but requires only minimal additional memory. Importantly, during inference, both the computational resources and parameter count of the Tourbillon version are identical to those of the original model.

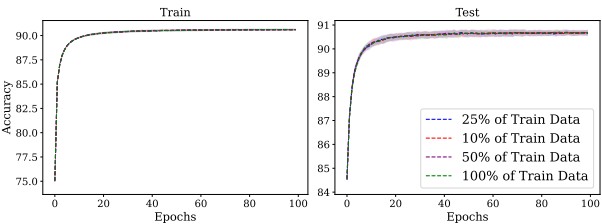

**Figure 6.** The train and test accuracy of four asynchronous Tourbillons with the use of 10%, 25%, 50%, and 100% of the labeled data. Each line corresponds to the mean of five distinct runs with the standard deviation shown as the shaded area.

The key idea is to substitute each sequential layer with a small CAE, enabling the encoding of intermediate data in a manner that facilitates its transmission to the higher layers of the network. This conversion algorithm is represented in Algorithm A.2. We use Algorithm A.2 to convert a U-Net [16] and a feed-forward architecture into their Tourbillon versions. The details of the training are given in the

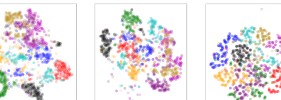 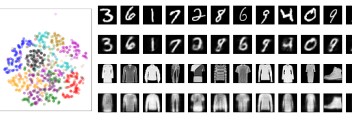

**Figure 7.** tSNE plots for the 2D visualization of 1000 random samples from the MNIST, Fashion MNIST, and CIFAR-10 datasets. Examples of reconstructed images from the MNIST and the Fashion MNIST datasets using Tourbillon are shown on the right.

Appendix. Table A.3 illustrates that the process of converting an architecture to its more biologically plausible Tourbillon version incurs a relatively small cost in terms of accuracy degradation (94% vs 99%).

**Table 4.** Comparison of networks trained with BP vs their Tourbillon version: U-Net numbers show mean-squared error, feed-forward numbers show classification error.

| U-Net | | |
|---|---|---|
| | MNIST | CIFAR-10 |
| BP Train | 0.0031 (5.8e-05) | 0.0024 (5.8e-05) |
| BP Test | 0.0031 (5.8e-05) | 0.0026 (e-04) |
| Tourbillon Train | 0.0111 (e-04) | 0.0733 (3.2e-04) |
| Tourbillon Test | 0.0113 (2e-04) | 0.0783 (3.2e-04) |
| **Fully Connected** | | |
| | MNIST | CIFAR-10 |
| BP Train | 1.04 (0.20) | - |
| BP Test | 1.18 (0.20) | - |
| Tourbillon Train | 5.21 (0.40) | - |
| Tourbillon Test | 5.82 (0.40) | - |

## 5 Conclusion

Tourbillon represents a systematic approach towards addressing the problems of deep learning in biological or neuromorphic networks. In essence, it is a stack of circular autoencoders trained by recirculation, hence the name Tourbillon associated with turbulence in French. Moreover, in horology, a tourbillon is an addition to the mechanics of a watch escapement to increase its accuracy. While we do not claim to have increased accuracy, we have shown that the Tourbillon approach shows similar performance to backpropagation, at least on MNIST, Fashion MNIST, and CIFAR-10.

Several issues have been identified that will require additional research. The first one is to study the scaling of the Tourbillon architecture so it can be trained on real-world data sets such as ImageNet [17]. The second one is to study whether local learning algorithms are necessary to also fine-tune the stack during regression or classification. For instance, using decoder weights as the random matrices of FA may be a possibility. The third one is the issue of convolutions that was only incrementally addressed here by showing that Tourbillon with convolutions does better than FA, but lags behind BP. And last, there is the study of Tourbillon with spiking neurons.

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

# A  Appendix

In this appendix, we provide additional details regarding the algorithms and the experiments. Each section in this appendix corresponds to the section with the same title in the main article. All the experiments are conducted using a single NVidia Titan X GPU.

## A.1  Biological Plausibility

We describe the learning equation (weight update) using post- and pre-synaptic terms. For a forward weight $W_i$ at layer $i$, the backpropagation learning equation can be written as:

$$\Delta W_i = \eta B_i^{\text{post}} H_{i-1}^{\text{pre}}, \quad B_L^{\text{post}} = T - H_L, \quad (4)$$
$$B_{i-1}^{\text{post}} = F'_{i-1} \circ W_i^T B_i^{\text{post}}$$

where $\eta$ denotes the learning rate, $B_i^{post}$ denotes the postsynaptic backpropagated error at layer $i$, and $H_{i-1}^{pre}$ denotes the pre-synaptic activity. $F'_{i-1}\circ$ denotes the component wise multiplication by the vector of activation function derivatives in layer $i-1$. $T$ and $H_L$ are the targets and the activation at the top layer $L$. Note that in order to avoid further cluttering the notation, we omit the transpose sign for all the presynatpic terms throughout this document.

### A.1.1  Feedback Alignment

Feedback Alignment (FA) or Random Backpropagation (RBP) refers to a family of algorithms [18, 19, 20, 21] that address the weights transport problem by using non-symmetric, and usually random, weights in the backward pass as follows:

$$\Delta W_i = \eta B_i^{post} H_{i-1}^{pre}, \quad B_L^{post} = T - H_L, \quad (5)$$
$$B_{i-1}^{post} = F'_{i-1} \circ R_i B_i^{post}$$

where $R_i$ denotes the random fixed matrices (random backward channels) to fix the issue of weights transport.

While FA and its variants address the weight transport problem, by themselves they do not address the other problems. Among several flavors of FA [19, 22], Direct Feedback Alignment (DFA) [20], backpropagates the error signal obtained at the top layer to each of the lower layers independently using direct fixed random matrices. By this means, DFA can address the issues of spatial locality and distance implausibilities. The learning equation of DFA can be written as follows:

$$\Delta W_i = \eta B_i^{post} H_{i-1}^{pre}, \quad B_i^{post} = R_i(T - H_L) \quad (6)$$

A version with component-wise multiplication by the derivatives of the corresponding activation functions is also possible. Experiments reported in the

literature suggest that FA and its variants do not work well with convolutional layers [23, 24, 25]. A few methods have been proposed to address this apparent weakness of FA algorithms, however, most of them introduce more constraints and dependence on the forward pass which may make them less biologically plausible [24, 26, 27].

### A.1.2  Difference Target Propagation

Difference Target Propagation (DTP) [5] as another biologically plausible model, trains the weights using a local target at each layer $\hat{Y}_i$ that is propagated from the original target $Y$ to each of the lower layers using learnable weights $G_i$.

$$\Delta W_i = \eta \hat{Y}_i^{post} H_{i-1}^{pre}, \quad \hat{Y}_i = \hat{Y}_{i+1} G_{i+1} \quad (7)$$

The $G_i$s are trained in the forward pass to approximate the inverse of the forward operation at each layer. Propagating the target using $G_i$s at the top two layers is dependent on the backpropagation and weight transport [25]. Also, the forward and backward passes through the network are completely clocked to learn $G_i$s. However, since the information flows through layers independently, the variables are local in space, thus, this architecture can address space-locality and distance implausibilities.

### A.1.3  Stacked Autoencoders

A well known approach to address the labeling issue is using a stack of autoencoders [28, 29], where each autoencoder learns to reproduce the hidden representation of the previous one in a self-supervised manner, allowing the stack to learn increasingly abstract representations without labels. Labels are only used to train the top layer in a supervised way, with the option to fine-tune all layers via backpropagation [30].

This approach also addresses the distance and developmental modularity issues since backpropagation within each autoencoder limits error gradients to short distances and allows training to begin before the entire architecture is complete. However, stacked autoencoders do not solve the locality and weight transport issues. Each autoencoder, being deep, requires backpropagation across at least two adaptive layers, necessitating a learning channel for error signals and symmetric weights to implement backpropagation.

### A.1.4  Forward Forward

The recently introduced Forward Forward algorithm (FF) [14], attempts to address the implausibility through a contrastive learning framework. Positive and negative data are fed through the network. Then the weights can be updated using a local target

defined at each layer as follows:

$$\Delta W_i = \eta(||H_i^+ - H_i^-||^2)^{post} H_{i-1}^{pre} \quad (8)$$

FF uses variables that are local in space and can be assumed to be local in time (due to the short neural distance). Given the contrastive learning framework, it can be trained in a self-supervised manner, however, the computation remains heavily clocked for feeding positive and negative data one at a time.

## A.2  Training Tourbillon CAEs

### A.2.1  Recurculation Learning Rules

We start by introducing two new learning rules (in addition to the main rule of recirculation which can be found in Equation 1).

$$\Delta W_i = \begin{cases} \eta(H_i^t - H_i^{t'})^{post}(H_{i-1}^t)^{pre}, & (a) \\ \eta(H_i^t - H_i^{t'})^{post}(H_{i-1}^{t'})^{pre}, & (b) \\ \eta(H_i^t - H_i^{t'})^{post}(H_i^t - H_{i-1}^{t'})^{pre} & (c) \end{cases} \quad (9)$$

Rule (a) is the main learning rule of recirculation. (b) is a slightly different version where the pre-synaptic term is computed at the later cycle ($t'$). In our experiments we observe that learning rule (b) provides trainability, however, it leads to a higher reconstruction error in a CAE across all datasets and architectures.

Given that rules (c) and (a) have been identified as the best-performing rules, our focus is on comparing these two rules. Specifically, we examine the characteristics of these rules in terms of their impact on training dynamics.

In Figure A.1, we observe the training process of a circular autoencoder using rule (c), which incorporates symmetric terms for both pre- and post-synaptic activation differences. The graph illustrates that the use of the symmetric learning rule leads to the convergence of the loss function during training. Moreover, compared to rule (a), this rule exhibits smoother training dynamics. The smoother training dynamics associated with rule (c) or the symmetric learning rule suggest that it promotes more stable and consistent updates to the model parameters, leading to improved training performance.

Due to the inherent cyclic structure of circular autoencoders, the data circulates through the model in multiple time steps where neurons produce different activations. Assuming that the time frame between two consecutive cycles is short, the difference between two consecutive activations of a neuron $[H_j^{t+1} - H_j^t]$ can be interpreted as the activation rate of that neuron at time $t$. Therefore, rule (c) be also seen as the multiplication of the post-synaptic and pre-synaptic neurons' activation rates. This implies that the connection between two neurons ($\Delta w_{ij}$)

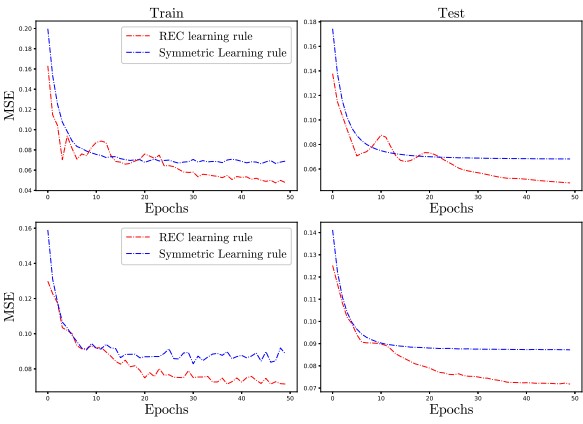

**Figure A.1.** The different behavior of the two recirculation learning rules (a) and (c). The REC learning rule corresponds to rule (a) and the symmetric learning rule corresponds to rule (c). While both demonstrate similar performance in training models using MNIST (first row) and Fashion MNIST (second row), the symmetric learning rule has smoother trajectories. All trajectories correspond to the best run out of three. The noise was removed for clarity.

must be strengthened (or weakened) if neurons' activation is correlated. This behavior closely matches the concept of spike time-dependent synaptic plasticity Hebbian (or anti-Hebbian) learning rules which was proposed using the temporal derivative of the activity of the post-synaptic neuron [13, 31] to encode error derivatives.

However, during training a circular autoencoder using rule (c), the model tends to be trapped into a mode-collapse state where the reconstructed images are the mean of the entire dataset. This mode collapse state can be seen in Figure A.2. Therefore, despite the interesting interpretation and intuition behind rule (c) with its symmetric terms, the problem of mode collapse confined our studies to the use of the main recirculation learning associated with rule (a).

### A.2.2 Training CAEs

Table A.1 summarizes the parameters used for training the CAEs in Section 4.1 of the main article. Additionally, all models were trained for 100 epochs with a batch size of 64. To optimize the activation function and learning rates, a grid search was conducted, resulting in the use of *tanh* activation function and a learning rate of 0.01 for the initial layers. Subsequently, a smaller learning rate of 0.001 was employed for the remaining fully connected layers across all architectures. For the models that incorporated convolutional layers and were trained using CIFAR-10, a learning rate of 0.001 was used for the initial layer, while a learning rate of 0.0001 was applied to the subsequent layers.

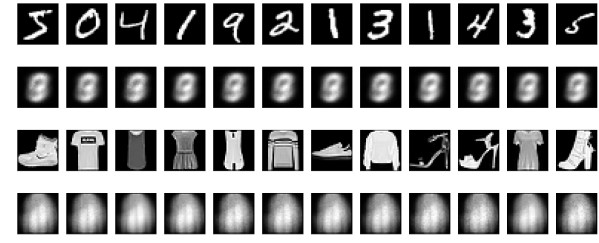

**Figure A.2.** Examples of reconstructed images from the MNIST and Fashion MINST datasets using the circular autoencoder trained with rule (c). There is a mode-collapse effect and the model reconstructs the mean of the data.

In the case of the convolutional models, zero padding is crucial to maintain the size of the input, ensuring that the spatial dimensions are preserved during the convolutions. The plots in the first row of Figure 5 display the training and test error curves for the convolutional circular autoencoder trained using the CIFAR-10 dataset. Notably, these experiments demonstrate that the use of recirculation leads to comparable reconstruction errors, and in some cases even superior, to those achieved with traditional backpropagation and random backpropagation techniques.

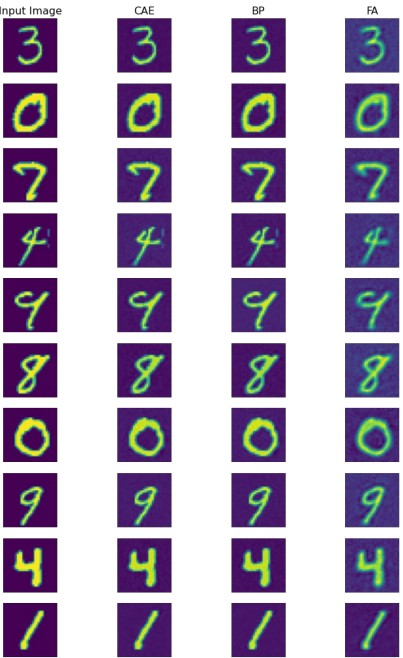

**Figure A.3.** Samples of reconstructed images from MNIST using autoencoders trained with BP, FA, and a CAE trained with recirculation.

**Table A.1.** CAE size refers to the number of hidden layers except for the input and output layers. For CIFAR-10, we only show the kernel size of the encoder part of the CAEs. The decoder of the CAEs used a symmetric kernel size.

| CAE size | MNIST and Fashion MNIST | CIFAR-10 | |
|---|---|---|---|
| | Hidden Layers Dim | Kernel | Stride |
| 1 | 784-256-784 | $(3 \times 5 \times 5 \times 3)$ | $(1 \times 1)$ |
| 3 | 784-256-64-256-784 | $(3 \times 5 \times 5 \times 3)$-$(3 \times 5 \times 5 \times 6)$ | $(1 \times 1)$ |

## A.3 Stacking CAEs With Various Depth and Training Algorithms

Here we explain the details of the experiments conducted in Section 4.2. Specifically, Table A.2 summarizes the parameters used for stacking and training the CAEs.

The size of the CAE architectures is explained in Table A.2. We use the same architectures for both sequential and asynchronous training algorithms. For the sequential training, each CAE is trained for 100 epochs with a batch size of 64. We use the learning rates explained in the previous section.

For the asynchronous training algorithm, we use a batch size of 64 and we train the entire stack for 3000 iterations. According to Algorithm A.1, an iteration refers to feeding one batch of data through the stack and updating the weights of one CAE within the stack. During our observations, we noted that when employing the same learning rate as the sequential training algorithm, the reconstruction loss exhibited a noisy trajectory with a limited convergence rate. To address this issue, we conducted several grid searches to identify an optimal approach. Our findings indicated that utilizing lower learning rates for the upper CAEs in the stack during the initial stages of training was crucial. By gradually increasing the learning rate of the upper CAEs, we observed a decrease in the reconstruction loss, eventually aligning all the CAEs in the stack to use the same learning rate. This adjusted learning rate strategy enabled more stable and efficient training, facilitating improved convergence of the reconstruction loss. We

provide a pseudocode of the asynchronous training in Algorithm A.1. To further enhance clarity, we have depicted the schematics of the asynchronous training algorithm in Figure A.4.

---
**Algorithm A.1** Asynchronous training
---
**Input:** $T$: A stack of $m$ sequential circular autoencoders $T = CAE_m \circ ... \circ CAE_1$, $CAE_i = D_i \circ E_i(datasample)$, $data$: training data, $S$: steps, $E_i$ and $D_i$ are the encoder and decoder of $CAE_i$
**for** $i = 1$ **to** $S$ **do**
$\quad 1 \le j = random \le m$
$\quad h = E_{j-1} \circ ... \circ E_1$
$\quad circulation(CAE_j, h)$
**end for**

---

To evaluate the performance of the Tourbillon architecture when adding the top classification layer, we conducted a comparison with similar architectures trained using backpropagation, FA, and DFA. Figure 4 presents the results of this experiment specifically for the Tourbillons, trained sequentially with a stack of three fully connected CAEs using the Fashion MNIST dataset. Additionally, the second row of Figure 5 showcases similar results obtained for the Tourbillons, trained sequentially with a stack of two convolutional CAEs using the CIFAR-10 dataset. All the parameters of the CAEs are explained above. In all cases, we can observe that Tourbillon can achieve comparable performance to models trained using BP or FA, showing the viability of Tourbillon while being the most biologically plausible model.

## A.4 Conversion to Tourbillon

As described in Section 4.4, we follow the Algorithm A.2 to build the biologically plausible version of the architecture for a U-Net autoencoder model that addresses all the problems mentioned in the Introduction. In addition, we also convert a feed-forward fully connected network architecture to its biologically plausible version. Results obtained on MNIST and Fashion MNIST are very similar, therefore we reported the results obtained on MNIST and CIFAR-10. The U-Net architecture for the MNIST dataset comprises a two-layer encoder and a two-layer decoder. The encoder layers compress the data from

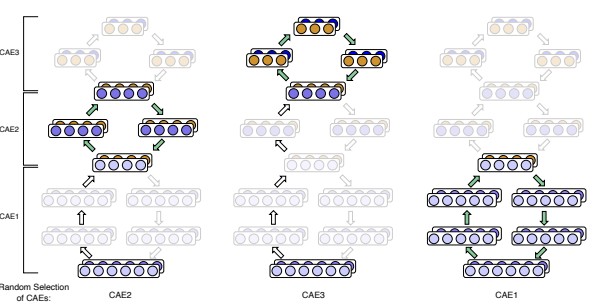

**Figure A.4.** Random phases of the asynchronous training. Each time, one CAE is selected randomly and trained by recirculating the information.

**Table A.2.** Depth refers to the number of CAEs used to construct the stack. For CIFAR-10, we only show the kernel size of the encoder part of the CAEs. The decoder of the CAEs used a symmetric kernel size.

| Depth | MNIST and Fashion MNIST Input and Hinge Layers Dim | CIFAR-10 | |
|---|---|---|---|
| | | Kernel | Stride |
| 2 | (784,256)-(256,128) | $(3 \times 5 \times 5 \times 3)$ | $(1 \times 1)$ |
| 3 | (784,256)-(256,128)-(128,64) | $(3 \times 5 \times 5 \times 3)$ | $(1 \times 1)$ |
| 4 | (784,256)-(256,128)-(128,64)-(64,64) | | |

---

**Algorithm A.2** Conversion to Tourbillon

**Input:** $M$: A network with $m$ sequential layers $L = [L_1, ..., L_m]$, $data$: training data.
**Output:** $M'$: The biological plausible version of the architecture.
Initialize $L' = []$.
**for** $i = 1$ **to** $m$ **do**
  **if** $L_i$ has learnable parameters **then**
    $l = train\_cae([L_1, ..., L_i])$
    $L'.append(l)$
  **else**
    $L'.append(L_i)$
  **end if**
**end for**
$M' = [data]$
**for** $i = 1$ **to** $m$ **do**
  $M' = L'_i(M')$
**end for**
**function** $train\_cae([L_1, ..., L_i])$:
$input = L_{i-1}(...(L_1(data)))$
$circular\_ae = G_i(L_i(input)), \quad G_i$:decoder
$recirculation(circular\_ae)$
return $L_i$
**endfunction**

---

**Table A.3.** Train/Test time and number of parameters used in the models. Models for MINST and Fashion MNIST are of identical sizes. Train times are the average time of one epoch. All times are in seconds.

| Model Name | Train/Test Time | Parameters |
|---|---|---|
| **Section 4.1** | | |
| BP (MNIST) | 3.95/411 | 403488 |
| FA (MNIST) | 4.05/420 | 403488 |
| CAE (MNIST) | 4.21/422 | 403488 |
| BP (CIFAR-10) | 2.40/231 | 244 |
| FA (CIFAR-10) | 2.44/231 | 244 |
| CAE (CIFAR-10) | 2.59/245 | 462 |
| **Section 4.2** | | |
| BP (MNIST) | 5.90/595 | 49220 |
| FA (MNIST) | 6.17/605 | 49220 |
| SAE (MNIST) | 6.20/620 | 981250 |
| Tourb (MNIST/seq) | 6.29/628 | 981250 |
| Tourb (MNIST/asynch) | 7.01/628 | 981250 |
| BP (CIFAR-10) | 3.66/350 | 720 |
| FA (CIFAR-10) | 3.66/350 | 720 |
| SAE (CIFAR-10) | 3.81/379 | 1380 |
| Tourb (CIFAR-10/seq) | 3.94/396 | 1380 |
| Tourb (CIFAR-10/asynch) | 4.19/420 | 1380 |
| **Section 4.3** | | |
| Tourb-10% (MNIST) | 7.01/628 | 981250 |
| Tourb-25% (MNIST) | 7.01/628 | 981250 |
| Tourb-50% (MNIST) | 7.01/628 | 981250 |
| Tourb-100% (MNIST) | 7.01/628 | 981250 |
| Tourb-10% (CIFAR-10) | 4.19/420 | 1380 |
| Tourb-25% (CIFAR-10) | 4.19/420 | 1380 |
| Tourb-50% (CIFAR-10) | 4.19/420 | 1380 |
| Tourb-100% (CIFAR-10) | 4.19/420 | 1380 |
| **Section 4.4** | | |
| U-Net (MNIST) | 5.12/520 | 327100 |
| Tourb-U-Net (MNIST) | 6.71/590 | 654200 |
| U-Net (CIFAR-10) | 3.35/377 | 630 |
| Tourb-U-Net (CIFAR-10) | 3.88/390 | 1260 |
| FC (MNIST) | 5.88/600 | 49220 |
| Tourb-FC (MNIST) | 7.01/628 | 981250 |

784 to 128, and then from 128 to 64. The decoder layers expand the data from 64 to 128, and then from 128 to 784. The U-Net architecture for the CIFAR-10 dataset comprises two 2D convolutional layers with kernels of size $(5 \times 5)$. For both the MNIST and CIFAR-10 U-Nets we use the same batch size, learning rate, number of epochs, and activation function used in the previous section. For the feed-forward architecture, we use a three-layer network with 256, 64, and 10 hidden units. For this architecture and its Tourbillon twin, we also use the same batch size, learning rate, number of epochs, and activation function as described in the previous section.

