# OpenReview forum: "Towards Biologically Plausible Learning By Stacking Circular Autoencoders"
_NLDL.org/2025/Conference — Submitted to NLDL 2025_

### Official Review · Reviewer_pejn · 2024-09-23
**Towards Biologically Plausible Learning by Stacking Circular Autoencoders**

**Confidence:** 4

**Summary:**

The paper introduces Tourbillon, a neural network architecture designed to tackle challenges in biological learning, like limited data and error propagation without symmetric connections. It uses circular autoencoders and recirculation algorithms in a self-supervised manner, with an optional classification layer. Initial tests on small/tiny datasets show that Tourbillon performs comparably to traditional backpropagation models and may outperform other biologically plausible approaches.

**Strengths:**

The paper introduces Tourbillon, a deep learning architecture specifically designed to address challenges associated with training neural networks in biological systems, such as spiking neural networks. Key issues include limited labeled data and difficulties with error signal propagation due to the absence of symmetric connections. Therefore, the authors used tourbillon to use circular autoencoders and utilize different recirculation algorithms in a self-supervised manner. It also includes an optional top layer for classification or regression tasks. The architecture is tailored to meet biological learning constraints, distinguishing it from traditional models that enhance existing engineering applications. Preliminary results from experiments with benchmark datasets like MNIST, Fashion MNIST, and CIFAR10 suggest that Tourbillon performs similarly to models trained with backpropagation and may even outperform other biologically plausible approaches. However, the novelty without proper comparison is not justifiable.

**Weaknesses:**

Major:

1-	The results are based on preliminary experiments with small benchmark datasets, which may limit the generalizability of the findings to more complex or real-world scenarios. Additionally, there is a concern regarding how the authors adapt continuous-form datasets for use with the biological plausibility concept. Typically, biological plausibility means SNN requires dataset preprocessing or spike train-based datasets. For example, the N-MNIST dataset is often used for training biological plausibility-based neural architecture. It would be useful to know how the authors handle this aspect.

2-	While the paper claims that Tourbillon performs comparably to models using backpropagation and may outperform other biologically plausible approaches, it lacks detailed comparisons and analyses of these models. This raises concerns about the validity of the authors' claims without thorough comparative evidence. The NLDL conference is reputable, and for a full publication, it is essential that the authors provide detailed results rather than a general explanation of the SNN methodology. Given the limited page constraints, the authors should focus on presenting their own findings in detail to justify their claims.

3-	How were the parameters for training the Circular Autoencoders (CAEs) determined? Are there specific reasons why certain parameter choices (e.g., CAE size, number of cycles) were preferred over others? The authors should provide the exact parametric values used in their experiments, as these details are very important for other researchers attempting to replicate or build upon their findings.

4-	The paper mentions various training dynamics, including different learning rules and their effects on reconstruction loss. Are there detailed comparisons and justifications for why certain rules performed better? How do these findings align with the goals of demonstrating biological plausibility?

5-	The authors demonstrate in the experiment section that recirculation methods achieved comparable or superior reconstruction errors compared to backpropagation and other methods. Are the comparisons thorough and statistically significant? How do the results hold up across different datasets and architectures?

6-	Section 4.2- The section notes that Tourbillon successfully captures crucial information of the input images. Can the authors provide more detail on the quality of these reconstructions? How do the reconstructed images compare to those produced by other methods in terms of fidelity and accuracy?

7-	Section 4.3- How does the conversion to a Tourbillon-like version affect the overall architecture and functionality of the original neural networks (e.g., U-Net and feed-forward architecture)? Are there any qualitative differences observed in the performance or behavior of the converted models?

8-	On page 5, right column, last paragraph- The authors discussed ImageNet as a real-world dataset that would be important for scaling the Tourbillon architecture in research. Why was ImageNet not used in the current study? What are the specific challenges or limitations that prevented its use, and how do the authors plan to address these issues in future work?

Minor Mistakes

●	Authors need to cite recent state-of-the-art research studies.

**Justification:**

The paper introduces the Tourbillon architecture, which aims to address challenges in training neural networks within biological systems using circular autoencoders and self-supervised learning. While this approach is innovative but didn’t show promise in preliminary experiments with benchmark datasets, the paper has significant limitations that undermine its contributions.
Firstly, the results are based on small benchmark datasets, which raises concerns about the generalizability of the findings to more complex or real-world scenarios. Additionally, the adaptation of continuous-form datasets to the concept of biological plausibility, typically requiring spike train-based datasets, is not adequately addressed.

The paper also lacks detailed comparative analyses with existing methods and specific parameter values used in training which makes it difficult to fully validate the authors' claims. Moreover, the discussion on training dynamics and learning rules is insufficiently detailed, and the quality of reconstructions needs further elaboration. The impact of converting existing architectures to Tourbillon-like versions is not clearly demonstrated, and the absence of real-world datasets like ImageNet, mentioned as important for future work, is unexplained.
Given these substantial issues, including insufficient comparative evidence and lack of detail in critical areas, the paper does not meet the standards required for publication at this stage.

---

### Official Review · Reviewer_PEcT · 2024-10-08
**Nearly biologically plausible stacked circular autoencoders**

**Confidence:** 4

**Summary:**

Artificial deep neural networks while they have demonstrated success across most machine learning applications, they lack the biological constrains biological neural networks face. Biological constraints such as limited examples, local synaptic or weight learning or directionality. To bridge the gap between those constraints and artificial deep networks, the authors introduce a new architecture with circular autoencoders as backbone stacked into a multilayer structure. This architecture and self-supervised training meet more biological constrains than existing methods. They test the model in three datasets providing competitive performance to alternative solutions.

**Strengths:**

The manuscript is presented clearly, and it is technically sound. The model was tested on three publicly available datasets used to benchmark machine learning models. They compared the performance results to existing solutions showing comparable results. The performed ablation studies to test mode components relevance. They also clearly address some of the limitations of the model.

**Weaknesses:**

Biological constraints are relevant for the study of biological systems, like the brain. However, the authors only benched marked their model using datasets that test engineer applications. Since the authors failed to provide for a clear advantage to alternative models (e.g. higher performance or lower data demands), the impact of the solution is limited. To fully assess the potential, one would have to validate the model against state-of-the-art solutions (not off-the-shelf methods like shown) to illustrate the technological advantages; or one could test it on biological data to drive academic insights. The authors mention the critical limitations of the model, and it would be important to include them in future work.

The readability of the figures must be improved. All models should be used for all comparisons and results.

**Justification:**

The research was performed adequately and provides a new solution to bringing together biological and artificial neural networks, with the associated implications to neuroscience and AI.

---

### Official Review · Reviewer_jicL · 2024-10-08
**Novel idea with potential**

**Confidence:** 4

**Summary:**

This research paper proposes a biologically plausible deep-learning architecture called Tourbillon, which is a stack of circular autoencoders trained using recirculation algorithms. Tourbillon aims to overcome the challenges of training deep neural networks as biologically plausible systems. The paper highlights the limitations of existing biologically plausible approaches like Feedback Alignment, Difference Target Propagation, and Stacked Autoencoders. It demonstrates how Tourbillon addresses these limitations through its novel architecture and training algorithms. The authors present preliminary experiments on MNIST, Fashion MNIST, and CIFAR10 datasets, showing that Tourbillon achieves comparable performance to models trained with backpropagation and potentially outperforms other approaches. The foundations of the architecture and experiments are sound, but many aspects of the benchmarking of a novel architecture have been neglected (parameter size, training resources, limitations)

**Strengths:**

- Fairly novel architectural design with unique claims, though a combination of different ideas has been employed.
- Extensive experiments, including variable depth and architectural designs, have been conducted.
- The explanations for obstacles and architecture are clear, with sufficient figures and tables.
- Clear comparisons in functionality of the proposed model with others using a table. Furthermore, the claims have been sufficiently substantiated by the experiments.
- Implementation and codes made available

**Weaknesses:**

- Experiments limited to small datasets.
- Though mentioned in different parts of the paper, there is no concise related works section, especially regarding previous works and citations in biological plausibility in NNs. There is a lack of mention of other optimization techniques, such as the Adjoint Sensitivity method in NDEs or many other DE Solvers suitable to different systems and conditions.
- Some terms or phrases like ‘recirculating activities,’ ‘postsynaptic,’ and ‘presynaptic,’ etc., could use more explicit definitions.
- Though the solution offered claims to address all the issues mentioned in the introduction, the offered explanation does not seem sufficient for some problems like:
    - symmetry of connections
    - forward non-linearities
    - Clocked Computation etc.
- The memory, time, and other aspects of training, performance, and inference aren’t mentioned in the paper.

.
Questions:
- The performance drop in the classification task for MNIST, one of the simpler tasks in vision, is concerning. Do you think applications on larger and more complicated data (which is available to biological systems) would be lacking more?
- By spike in a biological system, do you mean neurons communicate by electrical pulses? aren’t they analogous to the binary form of numbers (as in a series of pulses)?
- Does this architecture training have memory or any other advantages over the existing algorithms? Were there any limitations (resource-wise or general)  faced during the training or inference? (I am curious as to whether it is practically advantageous in any one aspect)
- Stacked AEs are the closest architecture (as far as intuition goes) among the mentioned baselines, so I don't see a performance or loss comparison between them. Is there any reason why it wasn't mentioned?
- Though the objective is to create biologically plausible systems, if the resulting architecture doesn’t have advantages over the existing one (even if they aren’t biologically plausible), what is the motivation to pursue these constraints? (personally curious)

**Final Rebuttal Confidence:**

4

**Final Rebuttal Justification:**

The authors’ response and revision have effectively addressed several areas that were initially lacking. e updates to performance parameters for training and inference, along with the comparison to stacked autoencoders, were particularly helpful in addressing my concerns. Despite improvements, scalability remains a concern. The model’s struggle with large datasets limits its real-world applicability, questioning the practicality of a biologically plausible approach if it lacks clear advantages. Addressing this seems essential to unlock any potential in brain-inspired learning where such large databases play a crucial role.

As previously noted, the paper introduces a novel approach to biological plausibility and reveals hidden potential through diverse experiments. I will, therefore, maintain my initial judgment.

**Justification:**

Though more experiments are needed to explore the extent of biological plausibility, the paper puts forward a fairly novel architecture combining the advantages of many. Hence, the decision.

---

### Official Review · Reviewer_sdnL · 2024-10-09
**Interesting proposal but with serious issues**

**Confidence:** 4

**Summary:**

In this manuscript, the authors introduce a self-supervised architecture consisting of hierarchically stacked autoencoders trained with a recirculation algorithm, a model which they argue can address a large number of the incompatibilities of (approximate) backpropagation with biological constraints.
The manuscript first surveys the issues in the implementation of backprop in biological networks, then presents the Tourbillon architecture as a combination of the recirculation idea with a modular hierarchy. Experiments on small-scale models are performed to explore variations of the learning algorithm, model composition, and comparing against backprop and feedback alignment.

**Strengths:**

**1) Good summary of backprop vs. biological architecture.**

The identification of the 8 reasons in sect. 1 is useful and goes beyond the most often cited issues, by including such points as clocked computation and developmental modularity.

**2) Interesting architecture proposed**

The Tourbillon architecture as a stack of circular autoencoders (each of which can constist of multiple layers) is to my knowledge novel and differs from existing proposals of bio-plausible models. It draws heavily on the recirculation idea of Hinton & McClelland (1988), who already speculated about hierarchical versions of their model, however here a concrete and trainable form is given to this idea. I found interesting about the model that unlike deep belief networks, which at first sight seem similar apart from the learning rule, the Tourbillon model does not seem to require recurrent equilibration of the network state across the hierarchy, making learning modular. Nonetheless, I had serious questions about the viability of the architecture, see weaknesses.

**3) Generally good and concise summary of other bio-plausible learning rules in the appendix**

Appendix sect. A.1 was instructive and good to read (except for A.1.2, see below).

**Weaknesses:**

## Major issues

**1) Text and figures do not seem to be consistent in several places, and some figures seem incorrect**

 -  line 258-261: Contrary to the text, in Table 2 more than 1 cycle seems to be (slightly) better.
 - Fig 3 and Fig A.3: Is it plausible that CAEs are better than BP? This seems surprising. Especially in Fig. A.3, why are the BP (train red) lines so unstable while CAE and feedback alignment are nicely smooth? This points to a problem with the learning rate or possibly an error with the colors in these plots (note also that it seems that red and green may have been interchanged in the test and train panels?)
 - line 290-291: How do I see this in Table 3? This only seems to be the case for the CIFAR-10 model.
 - Do tables 2 and 3 show the train or the test reconstruction loss?
 - lines 301-303: This is not what Fig 4 shows, here Tourbillon seems clearly less accurate, as would be expected since during self-supervised training the goal is reconstruction, not to preserve features which would be useful for classification.
 - Table 3: The numbers highlighted in bold are not the lowest numbers. Instead, it seems that using less CAEs in the stack is better - this would be consistent with issue 3) below. Also, the caption says "CAEs with different depths" while likely this refers to depth in the sense of the number of CAEs in the stack?

**2) The model and learning rule are not clearly defined**

The Tourbillon model is the main contribution of the manuscript, but section 3 does not give a clear and full definition of model and algorithm. Things which are not described, neither in the main text nor the appendix:
 - What the concrete weight update in the presence of multiple cycles of recirculation is.
 - What the procedure of reconstruction is at test time (Is it that the test sample is propagated forward to the top of the stack, then propagated down to the output? This would be quite different from the procedure at training time of the lower CAEs, and require discussion).
 - The choice and comparison of sequential and asynchronous training, and of learning rules b) and c) in addition to a) are not motivated.
 - What is the motivation of the authors to propose a doubly-deep model, where each of the stacked CAEs can itself be a deep network?
My subjective recommendation would be to give a clear and full definition in the main text and move some of the empirical discussion of sect.4 to the appendix instead.


**3) The upper CAEs may be detrimental for reconstruction if there is no noise**

My interpretation is that at test time, the uncorrupted input sample is propagated forward to the top of the model via the encoder channel, then propagated backward along the separate decoder channel to yield the reconstructed sample.
If this is the case, the lowest level CAE would decode the image from a hidden image representation which is not its own encoder representation, but instead a representation which only approximates its encoder representation (this is what the CAE above was trained to do) - therefore introducing an additional error. It is then to be expected that the test loss using this procedure is always worse than the test loss using only the lowest CAE in isolation, where the encoded image is handed directly to the decoder as during training. This would explain why in Table 3 the smaller depth of stack have better loss than deeper stacks, and this should also be true for depth 1.
This would likely not mean that stacks of more than one CAE are useless, since if the input image is corrupted by noise the more compressed representation of the upper layers could help in denoising and yield a better reconstruction. However, if correct some experiments may need to be redone, and this would be a relevant issue to discuss (in a sense, the model would have no mechanism to add details to its reconstruction based on the sensory input, as e.g. a U-net with horizontal connections can, or a hierarchical predictive coding network where a compromise beween bottom-up and top-down information is created).



## Minor issues

 - The Tourbillon still seems to require clocking of the propagation: During training only certain parts of the network are activated for propagation and recirculation happens only in one CAE at a time. Also, it seems that the propagation procedure is different at train and at test times, while a biological architecture would likely not to distinguish these.
 - point 5) Labeling, in sect.1: Self-supervised learning is also normally done by backpropagation, and this is explicitly compared to in the manuscript. This is therefore not a reason why BP is not compatible with biology.
 - An important point which I was missing from the discussion of biologically plausible learning is that biological networks typically deal with time-dependent input sequences. This is a complication which the proposed Tourbillon model does not take into account.


### Small questions and typos
 - line 160 (and several other places): The eqref goes to the redundant eq.7 in the appendix, not to eq.1 in the main text.
 - line 170-171: Why is recirculation in the top layer identical to BP?
 - The question of distance plausibility was not clear to me. E.g. line 242, why are CAEs with less hidden layers more plausible?
 - line 327-333: Is the resulting model tourbillon-like or exactly a tourbillon model?
 - Sect. 4.3 and Table 4: Does the converted U-net have horizontal connections between the compressive and expansive branches as the original model?
 - Fig. 5: The t-SNE plots show that the representation has not lost the class structure. But to conclude that the model has improved the clustering a comparison to the input representation would be needed. (But in general shape and distances in t-SNE visualizations are hard to interpret)
 - Refs [5] and [6] are redundant.
 - Appendix sect. A.1.2: The explanation and discussion in lines 570-579 was unclear to me.
 - eq. 7 (and 1): It is not explained why versions b) and c) could be interesting. Is a) proposed here or is it the existing recirculation rule?
 - lines 641-645: It was unclear to me why the difference between activations from two sequential passes can be interpreted as a rate. In my understanding STDP depends on the relative timing of spikes in addition to the spike rate. So maybe a comparison to (Anti-)Hebbian plasticity can be made but I did not see how spike-time dependence arises here.
 - In Alg. A.1, while likely referring to encoder and decoder, E and D are not defined
 - Alg. A.2 in the def of circular_ae, the inverse of $L_i$ may not be defined, and if it exists the output of the CAE would perfectly match the input.

**Justification:**

While an interesting proposal is made in the manuscript, currently there seem to be factual errors in the figures and text (weakness 1) and the model may behave differently than expected (weakness 3). Therefore I recommend to reject the current version of the manuscript, as long as these issues can not be clarified as misunderstandings or corrected.

---

### Meta-Review · Area_Chair_VCcH · 2024-11-01

**Recommendation:** Reject
**Confidence:** 4

**Metareview:**

This work introduces Tourbillon, a novel self-supervised stack of circular auto-encoders trained via recirculation algorithms. The authors propose this method to tackle core challenges in biologically inspired learning, particularly in error signal propagation across symmetric connections. The approach is innovative, and all reviewers express interest in its potential, noting that it is the first practical implementation of such a concept inspired by previous methods.

However, there are notable concerns regarding the robustness of the experimental evaluation. Review sdnL highlights issues related to the upper CAE reconstruction with noise, questioning the method’s alignment with biologically inspired learning principles—a concern also echoed in review pejn. Additionally, the work lacks significant comparative analyses that could better contextualise its advantages and limitations, thereby diminishing the rigor of its experimental foundation.

While the strengths and novelty of the approach are clear, there is strong consensus among reviewers that the current evaluation lacks the depth needed to establish the method’s generalisability, scalability, and practical viability. Although the authors provide some discussion on scalability challenges, further analysis would enhance confidence in the method’s robustness. Notably, several reviewers emphasise the need for performance or computational improvements to substantiate the method’s significance. In line with the reviewers’ feedback, I recommend that further revisions focus on strengthening the evaluation and providing a more comprehensive justification for the conclusions drawn.

**Suggested Changes To The Recommendation:**

3: I agree that the recommendation could be moved up

---

### Decision · Program_Chairs · 2024-11-06

Reject